# Development of Nomograms for Predicting Prognosis of Pancreatic Cancer after Pancreatectomy: A Multicenter Study

**DOI:** 10.3390/biomedicines10061341

**Published:** 2022-06-07

**Authors:** So Jeong Yoon, Boram Park, Jaewoo Kwon, Chang-Sup Lim, Yong Chan Shin, Woohyun Jung, Sang Hyun Shin, Jin Seok Heo, In Woong Han

**Affiliations:** 1Division of Hepatobiliary-Pancreatic Surgery, Department of Surgery, Samsung Medical Center, School of Medicine, Sungkyunkwan University, Seoul 06351, Korea; sojeong.yoon@samsung.com (S.J.Y.); surgeonssh@gmail.com (S.H.S.); jsheo.md@gmail.com (J.S.H.); 2Biomedical Statistics Center, Samsung Medical Center, Research Institute for Future Medicine, Seoul 06351, Korea; br7.park@samsung.com; 3Department of Surgery, School of Medicine, Kangbuk Samsung Hospital, Sungkyunkwan University, Seoul 03181, Korea; skunlvup@naver.com; 4Department of Surgery, Seoul Metropolitan Government-Seoul National University Boramae Medical Center, College of Medicine, Seoul National University, Seoul 07061, Korea; limcs7@gmail.com; 5Department of Surgery, College of Medicine, Ilsan Paik Hospital, Inje University, Goyang 10380, Korea; ycshindr@gmail.com; 6Department of Surgery, College of Medicine, Ajou University Hospital, Ajou University, Suwon 16499, Korea; jwh2636@gmail.com

**Keywords:** pancreatic cancer, prognosis, prediction platform, nomogram

## Abstract

Surgical resection is the only curative treatment for pancreatic ductal adenocarcinoma (PDAC). Currently, the TNM classification system is considered the standard for predicting prognosis after surgery. However, the prognostic accuracy of the system remains limited. This study aimed to develop new predictive nomograms for resected PDAC. The clinicopathological data of patients who underwent surgery for PDAC between 2006 and 2015 at five major institutions were retrospectively reviewed; 885 patients were included in the analysis. Cox regression analysis was performed to investigate prognostic factors for recurrence and survival, and statistically significant factors were used for creating nomograms. The nomogram for predicting recurrence-free survival included nine factors: sarcopenic obesity, elevated carbohydrate antigen 19–9, platelet-to-lymphocyte ratio, preoperatively-identified arterial abutment, estimated blood loss (EBL), tumor differentiation, size, lymph node ratio, and tumor necrosis. The nomogram for predicting overall survival included 10 variables: age, underlying liver disease, chronic kidney disease, preoperatively found portal vein invasion, portal vein resection, EBL, tumor differentiation, size, lymph node metastasis, and tumor necrosis. The time-dependent area under the receiver operating characteristic curve for both nomograms exceeded 0.70. Nomograms were developed for predicting survival after resection of PDAC, and the platforms showed fair predictive performance. These new comprehensive nomograms provide information on disease status and are useful for determining further treatment for PDAC patients.

## 1. Introduction

Pancreatic ductal adenocarcinoma (PDAC) is a highly fatal malignancy, with a poor overall survival rate of approximately 10% [1]. The only curative treatment is surgical resection. As reported in recent studies, five-year survival for patients with resected PDAC has increased up to 17% [2]. The most widely accepted method for predicting survival outcomes after surgery is the American Joint Committee on Cancer (AJCC) system, consisting of tumor extent (T stage), lymph node status (N stage), and distant metastasis (M stage) (TNM staging) [3]. However, even the most recent AJCC system showed limited prognostic accuracy in previous multi-center validation studies [4,5]. In addition, because this system was largely based on a Western population, the validity of the system for general population is unclear.

Numerous attempts have been made to identify prognostic factors for PDAC apart from several well-known factors such as TNM stage and carbohydrate antigen 19–9 (CA 19–9). For preoperative factors, inflammatory markers such as neutrophil-to-lymphocyte (NLR) ratio have been studied as prognosticators [6,7]. Additionally, the prognostic values of nutritional status including sarcopenia are under assessment [8]. Intraoperatively, estimated blood loss (EBL) and transfusion have been reported as independent risk factors for survival [9]. In terms of pathologic features, lymph node ratio (LNR) and resection margin status are considered potentially associated with outcomes of patients with resected PDAC [2,10,11].

Some recent studies tried to develop platforms for predicting the prognosis of PDAC. The authors proposed nomograms predicting recurrence or survival, and the C-indices ranged from 0.64 to 0.73 [12,13]. These nomograms could be useful in easily calculating the probability of survival, but up-to-date variables were not included in the analyses. In the present study, we aimed to develop a new comprehensive nomogram with more prognosticators, which could estimate the probability of survival more accurately in patients with PDAC after surgical resection.

## 2. Materials and Methods

### 2.1. Data Collection

From January 2006 to December 2015, 963 patients were diagnosed with PDAC and underwent pancreatectomy at five different centers: Samsung Medical Center, Kangbuk Samsung Hospital, Seoul National University Boramae Medical Center, Ilsan Paik Hospital, and Ajou University Hospital. Patients’ medical records, including data for recurrence, were retrospectively reviewed. Patients with missing values or lacking recurrence data were excluded. Finally, a total of 885 patients were included in the analyses. This study was approved by the Institutional Review Board of Samsung Medical Center (Seoul, Korea, approval number: 2020–11–133).

### 2.2. Clinical Variables for Analysis

The demographic data and preoperative laboratory results, including CA 19–9, were collected. NLR and platelet-to-lymphocyte ratio (PLR) were calculated and included in the analysis as inflammatory markers. Preoperative computed tomography (CT) scans taken within 2 weeks of surgery were used to evaluate and diagnose sarcopenia. The diagnostic cut-off values for sarcopenia were based on a cohort study from a Korean national institution [14]: skeletal muscle index (SMI = skeletal muscle area at L3/height^2^) < 50.18 cm^2^/m^2^ for males and SMI < 38.63 cm^2^/m^2^ for females. Sarcopenic obesity (SO) was defined as visceral fat area (VFA)/SMI ≥ 2.5 according to our previous study on the effect of SO in patients with pancreatic cancer [8,15]. The preoperative relationship between tumors and the following major vessels were also evaluated: common hepatic artery (CHA), superior mesenteric artery (SMA), portal vein (PV), and superior mesenteric vein (SMV).

In terms of operation-related factors, types of operation including combined vascular resection, EBL, and the need for red blood cell transfusions were reviewed. In the final pathology reports, the size of tumor, lymph node (LN) status, and resection margin status were reported. Revised R1 resection refers to the existence of tumor within 1 mm of the resection margins and was included in R1. LNR was calculated as the number of metastatic LNs divided by the number of harvested LNs.

The optimal cut-off points were selected using log-rank test statistics to maximize the difference between low and high groups for the following variable: NLR, PLR, tumor size, and EBL, and the cut-off values were determined to be 2, 90, 2 cm, and 500 mL, respectively.

Determining cancer recurrence was based on abdomino-pelvic CT scans and CA 19–9 levels during postoperative surveillance. An additional chest CT or positron emission tomography (PET) scan was performed to confirm recurrent tumors. Recurrence-free survival (RFS) was measured by the time from surgery to the date of recurrence or last follow-up. Overall survival (OS) was defined as the time between surgery and death from any cause, by 30 June 2021.

### 2.3. Statistical Analysis

The demographic and clinicopathological data were presented as mean ± standard deviation (SD) for continuous variables, and frequency with percentile for categorical variables. Cox regression analysis was performed to assess risk factors for RFS and OS, and hazard ratios (HRs) were reported with 95% confidence intervals (CIs). The final model was determined after applying the backward selection method for variables with *p* < 0.05 in the multivariable model. To assess the discrimination of model, the C-index (Harrell’s concordance statistic) and the receiver operating characteristic (ROC) curve with time-dependent area under the curve (AUC) were used. Standard error (SE) and 95% CIs were presented. The bootstrap procedure with the number of 2000 was employed for internal validation of prediction models. Based on the bootstrap, the bias-corrected (overfitting-corrected) C-indices were obtained, and calibration curves were plotted with predicted and observed values. The results were considered statistically significant when two-sided *p*-values were <0.05. All statistical analyses were conducted using SAS software, version 9.4 (SAS Institute Inc., Cary, NC, USA.) and R software, version 4.0.5 (R Project for Statistical Computing).

## 3. Results

The demographic and clinicopathological data of the development cohort are summarized in Table 1. The mean age of the patients at operation was 63.1 years, and 58.8% were male. There were 99 (11.2%) sarcopenic patients and 285 (32.2%) with SO. On preoperative imaging, 166 (18.8%) patients showed arterial abutment and 260 (29.4%) patients showed PV/SMV abutment or invasion. Pancreatoduodenectomy, left-sided pancreatectomy, and total pancreatectomy were performed in 585 (66.1%), 295 (33.3%), and 5 (0.6%) patients, respectively. R0 resection was accomplished in 671 (75.8%) patients.

Table 2 shows risk factor analysis for RFS. In multivariable analysis, SO (HR: 1.22, 95% CI: 1.03–1.44, *p =* 0.020), elevated CA 19–9 (HR: 1.28, 95% CI: 1.07–1.53, *p =* 0.006), PLR (HR: 1.34, 95% CI: 1.07–1.68, *p =* 0.010), and CHA/SMA abutment (HR: 1.31, 95% CI: 1.08–1.59, *p =* 0.007) were preoperative factors associated with RFS. In addition, EBL (HR: 1.51, 95% CI: 1.28–1.78, *p <* 0.001), moderate and poor differentiation (HR: 1.43, 95% CI: 1.07–1.90, *p =* 0.015, HR: 1.83, 95% CI: 1.35–2.49, *p <* 0.001, respectively), tumor size (HR: 1.67, 95% CI: 1.33–2.09, *p <* 0.001), LN ratio (HR: 1.14, 95% CI: 1.08–1.20, *p <* 0.001), and tumor necrosis (HR: 1.56, 95% CI: 1.27–1.93, *p <* 0.001) were related to RFS.

In risk factor analysis for OS (Table 3), age at operation (HR: 1.16, 95% CI: 1.07–1.25, *p <* 0.001), underlying liver and chronic kidney disease (HR: 1.58, 95% CI: 1.12–2.22, *p =* 0.009 and HR: 3.62, 95% CI: 1.58–8.28, *p =* 0.002, respectively), preoperative PV/SMV invasion (HR: 1.74, 95% CI: 1.32–2.29, *p <* 0.001), PV/SMV resection (HR: 1.38, 95% CI: 1.10–1.72, *p =* 0.005), and EBL (HR: 1.36, 95% CI: 1.16–1.59, *p <* 0.001) were clinically significant factors in multivariable analysis. In pathologic features, moderate and poor differentiation (HR: 1.34, 95% CI: 1.01–1.78, *p =* 0.040 and HR: 2.20, 95% CI: 1.64–2.96, *p <* 0.001, respectively), tumor size (HR: 1.53, 95% CI: 1.24–1.89, *p <* 0.001), LN metastasis (HR: 1.69, 95% CI: 1.42–1.99, *p <* 0.001), and tumor necrosis (HR: 1.67, 95% CI: 1.37–2.04, *p <* 0.001) were related factors.

New nomograms predicting probabilities of RFS and OS at 2 years and 5 years after surgery were established (Figure 1), using the variables identified from the multivariable Cox’s proportional hazard models: sarcopenic obesity; elevated carbohydrate antigen 19–9; platelet-to-lymphocyte ratio; preoperatively-identified arterial abutment; estimated blood loss (EBL); tumor differentiation, size, lymph node ratio, and tumor necrosis for RFS models; age at operation; underlying liver disease; chronic kidney disease; preoperatively found portal vein invasion; portal vein resection; EBL; and tumor differentiation, size, lymph node metastasis, and tumor necrosis for OS models. Points were assigned to each variable considering the HR from the Cox model. The time-dependent AUCs were as follows: 0.742 (95% CI: 0.703–0.781, SE: 0.020) and 0.765 (95% CI: 0.717–0.813, SE: 0.025) for two-year and five-year RFS model, 0.723 (95% CI: 0.686–0.760, SE: 0.019) and 0.732 (95% CI: 0.699–0.767, SE: 0.017) for two-year and five-year OS model. Calibration plots of the models through 2000 bootstrap resamples were constructed to show the agreement between the predicted and actual probabilities (Figure 2). The bias-corrected two-year and fove-year AUCs were 0.737 (95% CI: 0.701–0.773) and 0.760 (95% CI: 0.717–0.803) for the RFS model and 0.721 (95% CI: 0.688–0.754) and 0.729 (95% CI: 0.693–0.766) for the OS model, respectively. Lastly, a user-friendly website containing calculators was built, in which users can obtain the survival probabilities automatically calculated by simply filling in the blanks (Samsung Medical Center—All rights reserved, © 2022. URL: http://pdacprognosis.smchbp.org/, accessed on 30 March 2022) (Figure 3).

## 4. Discussion

In the present study, we proposed nomograms for predicting RFS and OS after resection of PDAC. These new nomograms included recently evaluated diverse risk factors as well as conventional factors, which could be easily obtained from patient perioperative data. The predictive power of the nomograms was fair, with the AUC values exceeding 0.70. Additionally, the web calculators based on the nomograms have been established, providing a user-friendly interface for potential users.

During the process of investigating risk factors for composing new nomograms, some novel prognostic factors showed association with survival of PDAC patients. Among preoperative variables, PLR was significantly associated with RFS. Reportedly, chronic inflammation plays an important role in carcinogenesis of various GI tract malignancies as well as in PDAC [16,17]. Several studies have proposed possible links between PDAC and numerous inflammatory parameters, but the findings were somewhat inconsistent. Elevated NLR was shown to predict a poor prognosis in some studies [6]; however, only PLR was a significant risk factor in other studies [7]. In our previous single institutional study, both NLR and PLR were associated with the OS of patients with resected PDAC [2]. In the present study based on multi-center data, only PLR was significantly associated with RFS. Other parameters, such as lymphocyte-to-monocyte ratio, advanced lung cancer inflammation index, and prognostic nutritional index, which are readily available hematologic markers, should also be investigated. Further studies are needed in which the utility of several inflammatory markers is evaluated and the appropriate cut-off values identified to predict recurrence or survival.

Another significant preoperative factor related to RFS was SO. Sarcopenia has been an important issue for the last decade because many reports have suggested its relation to various adverse outcomes such as poor physical performance and even mortality [18,19]. Particularly for patients undergoing surgery for GI tract malignancies, sarcopenia is a risk factor for postoperative complications or poor survival [8,20]. In this study, SO rather than sarcopenia was an independent risk factor for RFS. SO is an indicator that reflects excessive body fat mass as well as low muscle mass, of which the pathogenesis involves diverse factors including both lifestyle and medical factors [15]. SO is a potentially adjustable risk factor that should be taken into consideration. Although only preoperative status was measured in the present study, sarcopenia or SO can change postoperatively because body composition is altered due to physiologic stress from surgery. Further in-depth analysis on the relationship between SO and oncologic outcomes would be helpful to emphasize the importance of SO to patients undergoing surgery for PDAC. In terms of diagnosis, several organizations including the European Working Group for Sarcopenia in Older People (EWGSOP) and the Asian Working Group for Sarcopenia (AWGS) have suggested different diagnostic criteria for sarcopenia and SO. In addition, various diagnostic tools for sarcopenia are being used, mostly dual-energy X-ray absorptiometry (DXA) or bioelectrical impedance analysis (BIA). In our study, we used CT scans for measuring body composition considering that every patient with a scheduled surgery takes routine preoperative CT. Recently, handgrip strength or cross-sectional area of bicep brachii muscle were suggested as assessment tools for sarcopenia [14,21]. The diagnostic concordance between the various tools has not been investigated, which may have led to inconsistency among studies in which the role of sarcopenia or SO on postoperative outcomes was investigated. In our institution, a prospective study using the various above-mentioned tools is planned to identify optimal diagnostic methods and clinical implications of sarcopenia-related markers in patients with PDAC.

Among pathological features, an important prognostic factor for resected PDAC is LN status. The N stage in the current TNM staging system refers only to the number of metastatic LNs. In this regard, concerns exist in terms of stage migration because the number of metastatic nodes could be affected by the number of harvested nodes. In a previous study, the total number of LNs obtained was reported to have a strong influence on the survival of PDAC patients [22]. The authors proposed that a minimum of 15 nodes be resected to gain survival benefits. The International Study Group on Pancreatic Surgery (ISGPS) also suggested that at least 12 or 15 LNs should be retrieved for accurate staging [23]. Extended lymphadenectomy, particularly in pancreatoduodenectomy, is not recommended due to its morbidity without obvious survival benefits [24]. Consequently, LN ratio could be more feasible to reflect the nodal status. LNR was identified as a prognostic factor for resected PDAC in several previous studies [10,11], and these findings are consistent with the results of the present study, which showed LNR was an independent risk factor for RFS. However, because consensus has not been reached regarding the optimal cut-off value for LNR, a numerical value was applied in our nomogram. Many studies have suggested potentially relevant cut-off values; however, the median number of harvested LNs was inconsistent among the studies. To effectively utilize the nodal status of PDAC, further analytical approaches using well-organized pathological data should be considered.

Regarding the variables included in the final models, the components were different between the RFS model and the OS model. In the nomogram for predicting OS, age at operation as well as underlying liver and chronic kidney diseases are included. Because OS refers to all-cause mortality, the factors which can affect the performance of patients are potentially associated with OS. These factors might also have influenced the completion of adjuvant therapy after surgery. The results of the present study could be a basis for future studies investigating the clinical implications of common chronic diseases and related parameters in patients undergoing pancreatectomy for PDAC. Similarly, the oncologic or survival benefit of PV/SMV resection (PV/SMVR), mostly performed with PD, has been a topic of debate, and this study showed that PV/SMVR was associated with poor OS. This result may indicate that venous resection is accompanied by increased morbidity with no actual survival benefit, as demonstrated in several previous studies [25,26]. Conversely, in a recent study, propensity score matching analysis showed that PV/SMVR might be a reasonable option for patients with PV/SMV involvement to achieve survival outcomes comparable to patients without PV/SMV involvement without increasing severe postoperative complications [27]. Further investigation is necessary to verify the actual survival benefit and safety of PV/SMVR in patients undergoing pancreatectomy.

In terms of predictability, the AUC values of our nomograms were >0.7, surpassing the predictive power of some previous prediction systems. First, van Roessel et al. performed external validation of the recent TNM staging system using an international multicenter cohort [5]. The results indicated the possibility for further improvement with a C-index value of 0.57. An original nomogram study was conducted in 2004 with prospectively collected single-institutional data, and the nomogram was developed to predict disease-specific survival at 1, 2, and 3 years after surgery [12]. The bootstrap-corrected C-index was 0.64, showing better discriminative power than contemporary TNM staging, but still limited predictability compared with our nomograms. Meanwhile, in 2013, there was a pilot study proposing a prediction platform for resected PDAC based on artificial intelligence technique [28]. Although the number of the subjects was only 84, the possibility that artificial intelligence might offer better prediction than conventional Cox regression models was demonstrated. Many studies are being performed to compare the predictive ability between nomogram and machine learning techniques using a database of several types of cancer [29,30]. In our institution, a combinations of various analytical techniques is being investigated to improve the predictive platforms.

The present study has several limitations. First, the study was based on a retrospectively reviewed database collected from multiple institutions. There are many possible sources of heterogeneity, including differences in diagnostic protocols, perioperative management, surgical techniques, and physicians who reviewed the medical records. In addition, selection bias might have affected the study because some patients with insufficient data regarding recurrence were excluded. However, this study has several strengths. We used a large database with several up-to-date variables, such as inflammatory markers and sarcopenia. Some variables were identified as independent prognostic factors and were included in the new models. With these prognosticators, nomograms were proposed that provide intuitive visualization of the predictive power of each factor. Furthermore, using the nomograms, a website was constructed where potential users can easily calculate the survival probability of a patient. Notably, our new platforms showed fair predictive ability, with AUC values exceeding 0.7. The development of another platform using artificial intelligence, which could have an advantage over conventional statistics in terms of data imputation, is planned at our institution. We are also planning to perform external validation in order to verify the utility of the new platforms for the general population.

In conclusion, new nomograms predicting prognosis of PDAC after surgical resection were proposed. These new platforms could offer a patient considerable insight into the disease status and act as a reference for further treatment. A future study with more extensive data and artificial intelligence technique is planned to improve the predictive performance of the platforms.

## Figures and Tables

**Figure 1 biomedicines-10-01341-f001:**
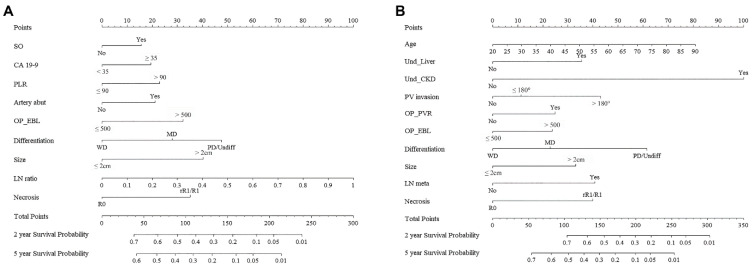
The new nomograms predicting recurrence-free survival (RFS) and overall survival (OS) of pancreatic cancer patients after surgery. (**A**) The nomogram for recurrence-free survival; (**B**) The nomogram for overall survival.

**Figure 2 biomedicines-10-01341-f002:**
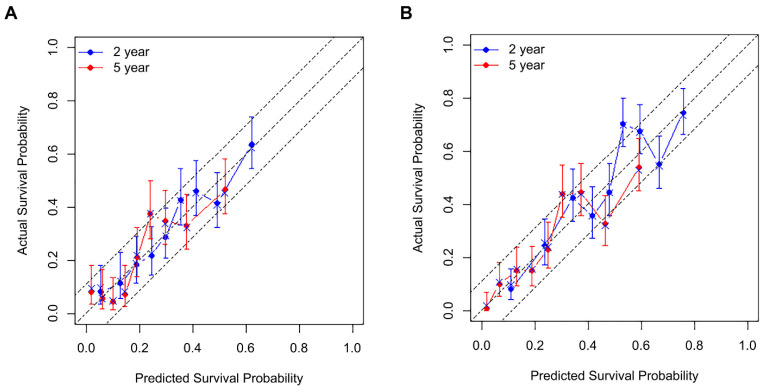
Calibration plots of internal validation for the new nomograms. (**A**) The plot of the nomogram predicting recurrence-free survival; (**B**) The plot of the nomogram predicting overall survival.

**Figure 3 biomedicines-10-01341-f003:**
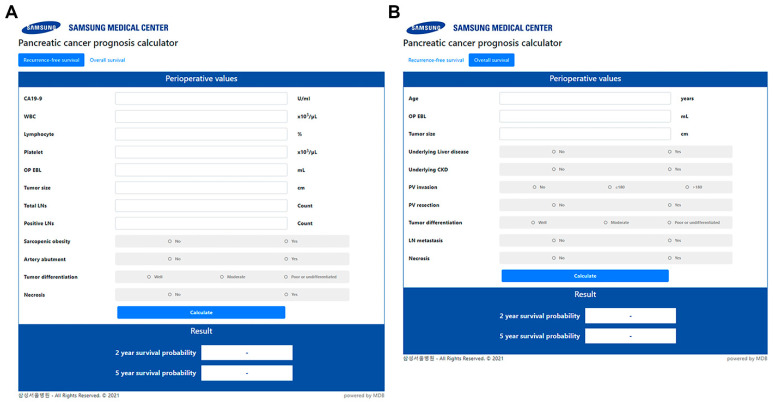
The website for calculating survival probability (http://pdacprognosis.smchbp.org, accessed on 30 March 2022). (**A**) The page for calculating recurrence-free survival; (**B**) The page for calculating overall survival.

**Table 1 biomedicines-10-01341-t001:** Demographic and clinicopathological data of the development cohort (*n* = 885).

Variable	N (%) or Mean (±SD)	Variable	N (%) or Mean (±SD)
Age at operation	63.1 (±10.3)	Operation	
Sex, male	520 (58.8%)	Pancreatoduodenectomy	585 (66.1%)
BMI	22.9 (±3.1)	Left-sided pancreatectomy	295 (33.3%)
ASA score		Total pancreatectomy	5 (0.6%)
I	176 (19.9%)	Combined PV/SMV resection	144 (16.3%)
II	632 (71.4%)	EBL > 500 mL	301 (34.0%)
III	77 (8.7%)	Intraop. RBC transfusion	149 (16.8%)
Underlying disease			
Cardiovascular	371 (41.9%)	Tumor differentiation	
Respiratory	66 (7.5%)	Well	85 (9.6%)
Liver	41 (4.6%)	Moderate	521 (58.9%)
CKD	6 (0.7%)	Poor/Undifferentiated	279 (31.5%)
DM	347 (39.2%)	Tumor size, cm	3.14 (±1.3)
Neoadjuvant treatment	33 (3.7%)	LN metastasis	562 (63.5%)
Sarcopenia, yes	99 (11.2%)	LN ratio	0.1 (±0.15)
Sarcopenic obesity, yes	285 (32.2%)	Tumor necrosis, yes	149 (16.8%)
Preop. elevated CA 19–9 ^a^	617 (69.7%)	R0 resection	671 (75.8%)
Preop. serum albumin	4.0 (±0.5)		
NLR > 2	451 (51.0%)	Length of stay, days	15.1 (±11.9)
PLR > 90	739 (83.5%)	Major complication, yes ^d^	158 (17.9%)
Preop. CHA/SMA abutment ^b^	166 (18.8%)	Adjuvant chemotherapy	508 (57.4%)
Preop. PV/SMV abutment ^b^	181 (20.5%)		
Preop. PV/SMV invasion ^c^	79 (8.9%)		

Abbreviations: N, number; SD, standard deviation; BMI, body mass index; ASA, American Society of Anesthesiologists; CKD, chronic kidney disease; DM, diabetes mellitus; Preop., preoperative; NLR, neutrophil-to-lymphocyte ratio; PLR, platelet-to-lymphocyte ratio; CHA, common hepatic artery; SMA, superior mesenteric artery; PV, portal vein; SMV, superior mesenteric vein; EBL, estimated blood loss; Intraop., intraoperative; RBC, red blood cell; LN, lymph node. ^a^ CA 19–9 ≥ 35 U/mL. ^b^ Abutment: abutment or contact of tumor ≤ 180°. ^c^ Invasion: contact of tumor > 180° or invasion. ^d^ Clavien–Dindo grade ≥ 3.

**Table 2 biomedicines-10-01341-t002:** Cox proportional hazard model for recurrence-free survival (*n* = 885).

Variable	Univariable Analysis			Multivariable Analysis		
	HR	95% CI	*p*	HR	95% CI	*p*
Age at operation	0.96	0.89–1.03	0.261			
BMI	0.99	0.97–1.02	0.560			
ASA score (ref. I)						
II	1.05	0.86–1.28	0.634			
III	0.89	0.63–1.24	0.479			
Underlying liver disease	1.28	0.89–1.83	0.186			
Underlying CKD	1.77	0.73–4.26	0.205			
Underlying DM	1.05	0.89–1.23	0.563			
Sarcopenia	1.03	0.80–1.32	0.821			
Sarcopenic obesity	1.22	1.03–1.43	0.020	1.22	1.03–1.44	0.020
Preop. Albumin	0.85	0.71–1.01	0.056			
Preop. elevated CA 19–9 ^a^	1.46	1.23–1.74	<0.001	1.28	1.07–1.53	0.006
NLR > 2	1.19	1.01–1.39	0.033			
PLR > 90	1.43	1.14–1.78	0.002	1.34	1.07–1.68	0.010
Preop. CHA/SMA abutment ^b^	1.30	1.07–1.58	0.007	1.31	1.08–1.59	0.007
Preop. PV/SMV abutment ^b^	1.07	0.88–1.31	0.475			
Preop. PV/SMV invasion ^c^	1.44	1.10–1.89	0.009			
Operation type (ref. PD)						
DP	0.88	0.74–1.04	0.132			
TP	1.83	0.76–4.42	0.181			
PV/SMV resection	1.22	0.99–1.51	0.062			
EBL > 500 mL	1.55	1.31–1.82	<0.001	1.51	1.28–1.78	<0.001
RBC transfusion	1.38	1.13–1.70	0.002			
Major complication (CD ≥ 3)	1.05	0.85–1.29	0.669			
Differentiation (ref. Well)						
Moderate	1.38	1.04–1.83	0.027	1.43	1.07–1.90	0.015
Poor/Undifferentiated	1.86	1.38–2.50	<0.001	1.83	1.35–2.49	<0.001
Tumor size > 2 cm	1.91	1.53–2.39	<0.001	1.67	1.33–2.09	<0.001
LN metastasis	1.53	1.30–1.81	<0.001			
LN ratio	1.16	1.10–1.22	<0.001	1.14	1.08–1.20	<0.001
Tumor necrosis	1.76	1.44–2.16	<0.001	1.56	1.27–1.93	<0.001
R1 resection (including rR1)	1.14	0.95–1.37	0.165			

Abbreviations: HR, hazard ratio; CI, confidence interval; P, *p*-value; BMI, body mass index; ASA, American Society of Anesthesiologists; CKD, chronic kidney disease; DM, diabetes mellitus; SO, sarcopenic obesity; Preop., preoperative; NLR, neutrophil-to-lymphocyte ratio; PLR, platelet-to-lymphocyte ratio; CHA, common hepatic artery; SMA, superior mesenteric artery; PV, portal vein; SMV, superior mesenteric vein; PD, pancreatoduodenectomy; EBL, estimated blood loss; Intraop., intraoperative; RBC, red blood cell; CD, Clavien–Dindo; LN, lymph node; LNR, lymph node ratio. ^a^ CA 19–9 ≥ 35 U/mL. ^b^ Abutment: abutment or contact of tumor ≤ 180°. ^c^ Invasion: contact of tumor > 180° or invasion.

**Table 3 biomedicines-10-01341-t003:** Cox proportional hazard model for overall survival (*n* = 885).

Variable	Univariable Analysis			Multivariable Analysis		
	HR	95% CI	*p*	HR	95% CI	*p*
Age at operation	1.07	0.99–1.15	0.088	1.16	1.07–1.25	<0.001
BMI	0.97	0.95–1.00	0.031			
ASA score (ref. I)						
II	1.19	0.98–1.44	0.076			
III	1.00	0.73–1.37	0.992			
Underlying liver disease	1.40	0.99–1.96	0.055	1.58	1.12–2.22	0.009
Underlying CKD	2.25	1.01–5.03	0.049	3.62	1.58–8.28	0.002
Underlying DM	1.17	1.01–1.37	0.040			
Sarcopenia	1.07	0.84–1.35	0.605			
Sarcopenic obesity	1.06	0.90–1.24	0.499			
Preop. Albumin	0.92	0.78–1.08	0.300			
Preop. elevated CA 19–9 ^a^	1.27	1.08–1.51	0.005			
NLR > 2	1.09	0.94–1.27	0.244			
PLR > 90	1.18	0.96–1.45	0.113			
Preop. CHA/SMA abutment ^b^	1.20	1.00–1.45	0.052			
Preop. PV/SMV abutment ^b^	1.25	1.04–1.51	0.016	1.16	0.95–1.41	0.152
Preop. PV/SMV invasion ^c^	1.99	1.56–2.56	< 0.001	1.74	1.32–2.29	<0.001
Operation type (ref. PD)						
DP	0.87	0.74–1.02	0.092			
TP	0.80	0.30–2.13	0.649			
PV/SMV resection	1.65	1.36–2.00	<0.001	1.38	1.10–1.72	0.005
EBL > 500 cc	1.38	1.18–1.61	<0.001	1.36	1.16–1.59	<0.001
RBC transfusion	1.09	0.89–1.33	0.421			
Major complication (CD ≥ 3)	1.06	0.88–1.29	0.536			
Differentiation (ref. Well)						
Moderate	1.42	1.08–1.88	0.013	1.34	1.01–1.78	0.040
Poor/Undifferentiated	2.34	1.75–3.12	<0.001	2.20	1.64–2.96	<0.001
Tumor size > 2 cm	1.73	1.40–2.13	<0.001	1.53	1.24–1.89	<0.001
LN metastasis	1.73	1.47–2.04	<0.001	1.69	1.42–1.99	<0.001
LN ratio	1.10	1.05–1.15	<0.001			
Tumor necrosis	1.90	1.57–2.30	<0.001	1.67	1.37–2.04	<0.001
R1 resection (including rR1)	1.20	1.01–1.43	0.038			

Abbreviations: HR, hazard ratio; CI, confidence interval; P, *p*-value; BMI, body mass index; ASA, American Society of Anesthesiologists; CKD, chronic kidney disease; DM, diabetes mellitus; SO, sarcopenic obesity; Preop., preoperative; NLR, neutrophil-to-lymphocyte ratio; PLR, platelet-to-lymphocyte ratio; CHA, common hepatic artery; SMA, superior mesenteric artery; PV, portal vein; SMV, superior mesenteric vein; PD, pancreatoduodenectomy; EBL, estimated blood loss; Intraop., intraoperative; RBC, red blood cell; CD, Clavien–Dindo; LN, lymph node; LNR, lymph node ratio. ^a^ CA 19–9 ≥ 35 U/mL. ^b^ Abutment: abutment or contact of tumor ≤ 180°. ^c^ Invasion: contact of tumor > 180° or invasion.

## Data Availability

Research data are not shared.

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
