# Peer review of "Development of Nomograms for Predicting Prognosis of Pancreatic Cancer after Pancreatectomy: A Multicenter Study"

_biomedicines, 2022, doi:10.3390/biomedicines10061341_

Round 1
Reviewer 1 Report
In the paper entitled “Development of nomograms for predicting prognosis of pancreatic cancer after pancreatectomy: A multicenter study”, the authors aimed to develop new predictive nomograms with more prognosticators, which could estimate the probability of survival more accurately in patients with pancreatic ductal adenocarcinoma (PDAC) after surgical resection. Indeed, to date, TNM classification system is considered the standard for predicting prognosis after surgery and the prognostic accuracy of the system remains limited. The new comprehensive nomograms would provide information on disease status and would be helpful for determining further treatment for PDAC patients.
This study may offer a further support to broaden the PDAC therapeutic treatment.
Data reported in this manuscript are interesting to open new perspectives for pancreatic ductal adenocarcinoma (PDAC) evaluation and treatment after surgical resection. The paper is well written. It can be considered for publicaton after the definition of a subset of analyzed data that will confirm the goodness of this study. I reported here my main concerns:
The authors should add an ABBREVIATIONS Section, since there are many acronyms without a reference to the complete name or the inverse. Some examples are “receiver operating characteristic, that means ROC, and acronyms that are difficult to find within the text. Please group them in a dedicated section and check for all the terms within the manuscript.
In the RESULTS Section, the authors have to reorganize the description of the “variable identified” and the way the graph in Figure 1 is constructed. It is a little bit confused and not easy to understandthe meaning of the graph by itself.
The manuscript is well written. However, data have to be used to analyze a larger cohort of patients in order to reducethe heterogeneity that also the authors reported as a limitation to the study.
In conclusion, this manuscript is really interesting and complete both in its content and in its conclusions. It can be improved by adding datasets analysis and enlarge the samples cohort. The experiments strongly support authors’idea and hypothesis
I think the paper meets the journal aims and can be considered for publication after the authors will respond to comments and eventually improve their analayzed data.
Author Response
Reviewer #1 comments:
In the paper entitled “Development of nomograms for predicting prognosis of pancreatic cancer after pancreatectomy: A multicenter study”, the authors aimed to develop new predictive nomograms with more prognosticators, which could estimate the probability of survival more accurately in patients with pancreatic ductal adenocarcinoma (PDAC) after surgical resection. Indeed, to date, TNM classification system is considered the standard for predicting prognosis after surgery and the prognostic accuracy of the system remains limited. The new comprehensive nomograms would provide information on disease status and would be helpful for determining further treatment for PDAC patients.
This study may offer a further support to broaden the PDAC therapeutic treatment.
Data reported in this manuscript are interesting to open new perspectives for pancreatic ductal adenocarcinoma (PDAC) evaluation and treatment after surgical resection. The paper is well written. It can be considered for publicaton after the definition of a subset of analyzed data that will confirm the goodness of this study. I reported here my main concerns:
- The authors should add an ABBREVIATIONS Section, since there are many acronyms without a reference to the complete name or the inverse. Some examples are “receiver operating characteristic, that means ROC, and acronyms that are difficult to find within the text. Please group them in a dedicated section and check for all the terms within the manuscript.
Response: Thank you for the advice. First, we added the term ROC in line 149. In accompany with that, standard error and 95% confidence intervals were presented.
We added an abbreviation section including major terms at the end of the abstract (from line 52).
- In the RESULTS Section, the authors have to reorganize the description of the “variable identified” and the way the graph in Figure 1 is constructed. It is a little bit confused and not easy to understand the meaning of the graph by itself.
Response: As you recommended, we referred back to the specific variables which were included in the nomograms, as follows (Line 184);
New nomograms predicting probabilities of RFS and OS at 2 years and 5 years after surgery were established (Figure 1), using the variables identified from the multivariable Cox’s proportional hazard models: sarcopenic obesity, elevated carbohydrate antigen 19-9, platelet-to-lymphocyte ratio, preoperatively-identified arterial abutment, estimated blood loss (EBL), tumor differentiation, size, lymph node ratio, and tumor necrosis for RFS models; age at operation, underlying liver disease, chronic kidney disease, preoperatively found portal vein invasion, portal vein resection, EBL, tumor differentiation, size, lymph node metastasis, and tumor necrosis for OS models.
The manuscript is well written. However, data have to be used to analyze a larger cohort of patients in order to reduce the heterogeneity that also the authors reported as a limitation to the study.
In conclusion, this manuscript is really interesting and complete both in its content and in its conclusions. It can be improved by adding datasets analysis and enlarge the samples cohort. The experiments strongly support authors’ idea and hypothesis
I think the paper meets the journal aims and can be considered for publication after the authors will respond to comments and eventually improve their analayzed data.
Reviewer 2 Report
I believe the paper is well written. I always worry about the exact use for a nomogram, especially one that is not likely to change treatment options. Though somewhat prognostic, it will not change the administration of adjuvant therapy or decision to proceed with or without surgery. Thus, I am not sure it will be widely used.
Author Response
I believe the paper is well written. I always worry about the exact use for a nomogram, especially one that is not likely to change treatment options. Though somewhat prognostic, it will not change the administration of adjuvant therapy or decision to proceed with or without surgery. Thus, I am not sure it will be widely used.
Response: We deeply appreciate your comment.
We, too, well recognize the limitations of the present study. As we mentioned, our major aim would be offering a patient and medical staff the insight on disease status by predicting prognosis.
Nevertheless, there are some factors which are potentially modifiable, such as sarcopenia. Also, we are planning a big-data investigation with many other patient-related factors as well as disease related factors in order to develop a more improved platform, and finally, in order to give helpful information for planning adjuvant treatment.
Thank you.
Round 2
Reviewer 1 Report
In the 2nd submission of the paper entitled “Development of nomograms for predicting prognosis of pancreatic cancer after pancreatectomy: A multicenter study”, the authors exhaustively responded to my comments.
With respect to my comments, I think they completely responded and added interesting and clarifying images to the manuscript, and the paper results clearer than in the first version.
The conclusion reported by the authors in each paragraph are clear and functional to the experiments reported, and strongly support the conclusions of the authors.
I think the paper meets the journal aims and can be considered for publication.